# Rapid Multiplex Real-Time PCR Method for the Detection and Quantification of Selected Cariogenic and Periodontal Bacteria

**DOI:** 10.3390/diagnostics10010008

**Published:** 2019-12-22

**Authors:** Jan Lochman, Martina Zapletalova, Hana Poskerova, Lydie Izakovicova Holla, Petra Borilova Linhartova

**Affiliations:** 1Department of Biochemistry, Faculty of Science, Masaryk University, Kotlarska 2, 611 37 Brno, Czech Republic; jlochman@seznam.cz (J.L.); 357594@mail.muni.cz (M.Z.); 2Department of Pathophysiology, Faculty of Medicine, Masaryk University, Kamenice 5, 625 00 Brno, Czech Republic; 3Clinic of Stomatology, Institution Shared with St. Anne’s Faculty Hospital, Faculty of Medicine, Masaryk University, Pekarska 53, 656 91 Brno, Czech Republic; hana.poskerova@fnusa.cz (H.P.); holla@med.muni.cz (L.I.H.)

**Keywords:** dental caries, periodontitis, dysbiosis, oral microbiome, risk factors determination

## Abstract

Dental caries and periodontal diseases are associated with a shift from symbiotic microbiota to dysbiosis. The aim of our study was to develop a rapid, sensitive, and economical method for the identification and quantification of selected cariogenic and periodontal oral bacteria. Original protocols were designed for three real-time multiplex PCR assays to detect and quantify the ratio of 10 bacterial species associated with dental caries (“cariogenic” complex) or periodontal diseases (red complex, orange complex, and *Aggregatibacter actinomycetemcomitans*). A total number of 60 samples from 30 children aged 2–6 years with severe early childhood caries and gingivitis were tested. In multiplex assays, the quantification of total bacterial (TB) content for cariogenic bacteria and red complex to eliminate differences in quantities caused by specimen collection was included. The mean counts for the TB load and that of ten evaluated specimens corresponded to previously published results. We found a significant difference between the microbial compositions obtained from the area of control and the affected teeth (*p* < 0.05). Based on this comprehensive microbiological examination, the risk of dental caries or periodontal inflammation may be determined. The test could also be used as a tool for behavioral intervention and thus prevention of the above-mentioned diseases.

## 1. Introduction

Dental caries and periodontal disease belong to the infectious diseases which represent the most common pathological conditions in the oral cavity. Dental caries is related to the destruction of the hard tissues of the tooth, while periodontal disease affects the soft tissues and alveolar bone that support the teeth. Although gingivitis is a reversible condition, affecting only the gingiva, it is a precursor of periodontitis. In children with primary dentition, early childhood caries (ECC) is a highly prevalent disease [1]. The definition of severe ECC (sECC) is any sign of smooth-surface caries in a child younger than three years of age, and from ages three through five, one or more cavitated, missing (due to caries), or filled smooth surfaces in the primary maxillary anterior teeth or a decayed, missing, or filled score of greater than or equal to four (age 3), greater than or equal to five (age 4), or greater than or equal to six (age 5) [2]. In addition, children with sECC had more dental plaque and gingival inflammation than caries-free children [3].

These diseases are associated with a shift from a symbiotic microbiota to dysbiosis, and an increase in the complexity of the oral microbiome [4,5,6]. Their progression involves multiple microbial interactions driven by different stressors and the responsiveness of the host’s immune system [7]. The concept of infectogenomics highlights the importance of host genetic factors in determining the composition of human microbial biofilms and the response to this microbial challenge [8].

A health-associated biofilm includes genera such as Neisseria, Streptococcus, Actinomyces, Veillonella, and Granulicatella [9]. There is evidence that a number of oral bacteria, like *Streptococcus mutans*, *Actinomyces* spp., and *Lactobacillus* spp., which are the components of the normal microbial flora, are positively related to decayed, missing, and filled teeth (DMFT) index [10]. Previous studies have implicated specific bacteria in the development and progression of periodontal diseases. Three of the most pathogenic periodontal species, *Tannerella forsythia*, *Treponema denticola*, and *Porphyromonas gingivalis*, comprise the group known as the “red” complex. The other species associated with periodontal inflammation, *Prevotella intermedia*, *Fusobacterium nucleatum*, and *Parvimonas micra* [11] are a part of the “orange” complex. In addition, Herbert et al. [12] suggested the periodontal bacteria *Aggregatibacter actinomycetemcomitans* as a potent immunoregulator of the host’s periodontal defense system and alveolar bone homeostasis. The microbial community of smoking-associated periodontitis is less diverse and distinct from that of non-smokers, indicating that smoking has an influence on periodontal ecology [6].

Currently, the methods for the detection of periodontal and cariogenic bacteria are predominantly semi-quantitative and based on cultivation technics on different selective media, immunochromatic detection of bacterial antigen or detection of specific DNA sequences using different methods including checkerboard hybridization, polymerase chain reaction (PCR), or real-time PCR. Studies of the microbiome associated with dental caries have largely relied on 16S rRNA sequence analysis, which is associated with PCR biases, low taxonomic resolution, and an inability to accurately study functions. Alternatively, shotgun whole metagenome sequencing does not involve gene amplification by PCR and allows the identification of the microbial taxa of the community to a higher resolution than 16S rRNA-based sequencing [13]. Unfortunately, a metagenomic approach is uneconomical for routine bacterial classification, where the challenge is rapid diversity analysis.

There is an assumption that samples from children with sECC and gingivitis may be appropriate for the testing of a new sensitive method for oral bacterial screening. The main aim of this study was to develop a tool for specific microbial analysis with high sensitivity, which could be used for the testing of subgingival and dental plaque samples from both adults and children. This method should be not only for the detection of bacteria but also for their quantification. The effort is to make the method economical and rapid with a high reproducibility rate.

## 2. Material and Methods

### 2.1. Study Population

Children from the South Moravia region of the Czech Republic were selected for this study according to the status of their dentition in 2018. The inclusion criteria for participants were preschool age (2–6 years), at least 16 primary teeth, general good health, and willingness of the parents or legal guardians to enter their children into the study. The exclusion criteria were the presence of one or more permanent teeth, a familial relationship between children, and ethnicity other than Czech Caucasian.

The study was approved by the Committee for Ethics of the Faculty of Medicine, Masaryk University, and St. Anne’s Faculty Hospital, Brno (1G/2017, 24 June 2016). Informed consent was obtained from all parents or legal guardians of the children prior to their inclusion in the study in line with the Helsinki declaration.

### 2.2. Clinical Data Collection and Sampling

Parents of the children were told that the child cannot eat, perform oral hygiene, chew gum, or drink (except still water) for 1 h before the dental examination and sample collection.

Thirty Czech children aged 2–6 years with sECC and gingivitis were selected and included in this study. For testing, subgingival and dental plaque samples (total 60) from two teeth (control tooth and affected tooth) of each child were used. Children were examined at the Ambulance of Pediatric Dentistry, University Hospital Brno, Czech Republic by an experienced pediatric dentist. The conditions of the clinical examination were published previously [14]. The decayed, missing, or filled teeth (d_3_mft) index was calculated using dental caries (d_3_ level) as a cut-off point for the detection of decay. The full mouth gingival status assessment was carried out visually before the collection of samples according to the WHO [15] based on the presence of clinical signs of inflammation (redness, swelling or hyperplasia). Only children with sECC (d_3_mft ≥ 6) and with gingivitis were included in this study. 

Subgingival and dental plaque samples from two teeth were collected. A tooth with localization 73 or 83 without dental caries and/or gingivitis in its surroundings was considered as a “control”, a tooth affected by dental caries and with gingivitis in its surroundings was selected as an “affected tooth”. Sampling was performed using a paper cone (ISO 40) and sterile tweezers, each tooth was encircled by the cone until it was covered with up to 2/3 biological material. Each paper cone was separately placed using tweezers into a sterile tube and stored at −20 °C.

### 2.3. Bacterial Strains

Bacterial strains *S. mutans* (DSM-20523), *L. acidophilus* (DSM-20079), *A. odontolyticus* (DSM-19120*), A. actinomycetemcomitans* (DSM-8324), *P. gingivalis* (DSM-20709), *T. forsythia* (DSM-102835), *T. denticola* (DSM-14222), *P. micra* (DSM-20468), *P. intermedia* (DSM-20706), and *F. nucleatum* (DSM-19679), used as positive controls to test the specificities of the primers and method performance, were obtained from the DSMZ-German Collection of Microorganisms and Cell Cultures (Braunschweig, Germany).

### 2.4. Isolation of gDNA from Bacterial Cultures, PCR Amplification of Specific DNA Targets and Cloning

Genomic DNA (gDNA) was isolated from cultured bacterial cells using QIAamp DNA Mini Kit (Qiagen, Hilden, Germany) according to the manufacturer’s recommendations. Oligonucleotides used in this study (Table 1) for the amplification of cariogenic bacteria (*S. mutans, Lactobacillus* spp., and *Actinomyces* spp.), periodontal bacteria (*A. actinomycetemcomitans*, *P. gingivalis, T. forsythia, T. denticola, P. micra, P. intermedia*, and *F. nucleatum*) and for the determination of total bacteria (TB) content in the samples were based on published sequences or designed de novo using Primer3 (v. 0.4.0) [16] and OligoAnalyzer software (Integrated DNA Technologies, Inc., Coralville, IA, USA) [17] (Table 1). 

Specific forward and reverse oligonucleotides (Table 1) were used to amplify 100–232 bp fragments from the target regions for each of the bacterial species (Table 1), and the amplified DNA fragments were cloned into the cloning vector pGEM^®^-TEasy (Promega, Madison, WI, USA). The correctness of these cloned DNA fragments was verified by sequencing using T7FW and M13R primers (data not shown).

### 2.5. Optimization of Multiplex Real-Time PCR Conditions

The double quenched TaqMan ZEN^®^ probes used in this study with reporter dyes FAM and HEX obtained from IDT (Integrated DNA Technologies, Inc., Coralville, IA, USA), and TaqMan probes with reporter dyes Cy5 and TexasRed obtained from Merck (Sigma-Aldrich Ltd., Gillingham, UK) are shown in Table 1. The individual TaqMan probes were initially tested in singleplex assays using the LightCycler^®^ 480 Instrument II (Roche Diagnostics AG, Rotkreuz, Switzerland). These assays were carried out in a total reaction volume of 25μL consisting of 12.5 μL QuantiFast Multiplex PCR Master Mix (Qiagen, Hilden, Germany), 160 nM of each primer, 200 nM of each probe and 2 μL of the target gDNA isolated from the specific bacterial culture (Table 1). The PCR amplification conditions were as follows: 5 min at 95 °C, 30 cycles at 95 °C for 45 s and 60 °C for 45 s with signal collection. 

Consequently, three quadruplex PCR assays were carried out on the LightCycler^®^ 480 Instrument II (Roche Diagnostics AG, Rotkreuz, Switzerland). The first assay contained the primers and probes for the amplification of cariogenic bacteria (*S. mutans, Lactobacillus* spp., and *Actinomyces* spp.) together with primers and probes for the determination of TB content in the samples (Table 1). Within the second assay, the primers and probes for the amplification of periodontal bacteria belonging to the orange complex (*P. micra*, *P. intermedia*, and *F. nucleatum*) were included together with those for the amplification of *A. actinomycetemcomitans* (Table 1). The final assay contained primers and probes for the amplification of those periodontal bacteria belonging to the red complex (*T. forsythia*, *T. denticola,* and *P. gingivalis*) together with primers and probes for the determination of TB content in the sample (Table 1). Individual multiplex real-time PCR assays were carried out in a total reaction volume of 25 μL consisting of 12.5 μL QuantiFast multiplex PCR kit (Qiagen, Hilden, Germany) 160 nM of each primer for cariogenic and periodontal bacteria and 80 nM of each primer for TB, 200 nM of each probe and 2 μL of the target gDNA isolated from the specific bacterial culture (Table 1). The final PCR amplification conditions were as follows: 5 min at 95 °C, 5 cycles at 95 °C for 45 s, 55 °C for 20 s, and 65 °C for 25 s without signal collection, and 30 cycles at 95 °C for 45 s and 60 °C for 45 s with signal collection. All amplifications and detections were carried out in the LightCycler^®^ 480 Instrument II (Roche Diagnostics AG, Rotkreuz, Switzerland) 96-well white reaction plate with optical sealing tapes (Bioplastics, Landgraaf, The Netherland). Data were analyzed using the LightCycler^®^ 480 software v1.5. 

### 2.6. Sensitivity of Multiplex Real-Time PCR Reactions

A ten-fold dilution series was prepared for each plasmid DNA starting from 1 × 10^8^ to 1 × 10^2^ copies/μL and used as a template in the real-time PCR assays under optimized conditions to determine amplification efficiency, reproducibility, and sensitivity. The limit of detection (LOD) was determined by PROBIT regression analysis at a 90% confidence level when six replicates of each plasmid DNA dilution were assayed per run. 

### 2.7. Validation of Method

The specificity of multiplex assays for individual bacteria detection in samples was verified by using the previously published protocol for real-time PCR assays [6]. The results obtained by both methods were compared.

### 2.8. Sequencing Analysis

Selected samples were tested to avoid false-positive results of the method. The PCRs were performed with the appropriate primers and the RED*Taq*^®^ DNA polymerase (VWR, PA, USA) in a 20 µL reaction volume. The conditions for PCR were: 95 °C for 3 min activation/denaturation step, followed by 40 cycles of 95 °C for 20 s, 60 °C for 20 s, 72 °C for 40 s, with a final extension for 5 min. The amplicons were purified by Exo-SAP treatment (ThermoFisher Scientific™, Waltham, MA, USA). Mixtures were incubated at 37 °C for 15 min and at 85 °C for 15 min to inactivate the enzymes, followed by sequencing with BigDye Terminator v3.1 kit (ThermoFisher Scientific™, Waltham, MA, USA). Sequencing reactions were purified by EDTA/ethanol precipitation, resuspended in 15 µl Hi-Di Formamide (ThermoFisher Scientific™, Waltham, MA, USA), and sequenced on an automated ABI3500 Genetic Analyzer (Life Technologies™, Carlsbad, CA, USA).

### 2.9. Statistical Analysis

All statistical analyses and associated plots were performed using the XLSTAT Software (Addinsoft Inc., New York, NY, USA). Pearson’s chi-square test was applied to test the differences in the distribution of negative, weakly positive and positive samples between the inflammatory and healthy sites. The two Student’s t-test between the percentage content of bacteria at inflammatory and healthy sites was used at the level of statistical significance of *p* < 0.05.

## 3. Results

### 3.1. Demographic Description and Clinical Data

In this study, 15 boys (4.4 ± 1.3 years, mean age ± standard deviation, SD) and 15 girls (4.3 ± 1.0 years, mean age ± SD) with 10.4 ± 4.0 mean d_3_mft ± SD (9.0 [7.0–13.0 and 6.0–20.0] median [interquartile range, IQR, and minimum–maximum]) were included. In total, 60 samples of dental and subgingival plaque were collected. Among the main affected teeth selected for sampling were the mandibular and maxillar molars in 56.6% and the incisors of the upper jaw in 26.7%.

### 3.2. Multiplex Real-Time PCR Conditions

The efficiency of the de-novo designed oligonucleotides, intended to be used in a multiplex reaction, was initially assessed in singleplex real-time PCR amplification using gDNA from corresponding bacteria species as a template. In these assays, the TaqMan probe of a concentration of 200 nM was determined to provide the highest RFU values representing the amount of amplified DNA. Consequently, different concentrations of primers (80–200 nM concentration) were tested in individual quadruplex assays. Based on the measured results, a primer concentration of 160 nM was determined to provide the best result for all the bacteria being analyzed, with the exception of TB, for which a concentration of 80 nM provided the best results. Finally, we also investigated various annealing temperatures to determine the optimum annealing conditions for the most efficient amplification for all bacterial targets in individual quadruplex assays (data not shown). The final optimal thermal cycling conditions included an incubation step for 5 min at 95 °C, five cycles at 95 °C for 45 s, 55 °C for 20 s, and 65 °C for 25 s without fluorescent detection (due to fluorescence signal fluctuation), and 30 cycles at 95 °C for 45 s and 60 °C for 45 s with fluorescent detection.

Subsequently, the amplification efficiency and limit of detection (LOD) were investigated using the optimized concentrations of oligonucleotides and the amplification profile. In these assays, we included a mixture of plasmids carrying bacterial templates DNA representing all four bacteria of interest in individual multiplexes. The amplification efficiency ranged from 0.93 to 1.16, where R^2^ calibration fit values close to 1.0 represent highly consistent and reproducible assays (Table 2). The average LOD of the multiplex PCR assay, as evaluated by PROBIT analysis, was about 10^3^ copies in reaction. Although the LODs for *T. denticola* and *Lactobacillus* spp. are a little lower (Table 2), in bacterial gDNA the target 16S region is present in multiple copies and the average concentrations of these bacteria in samples were usually higher (Figure 1). Based on all these results, we decided to use these multiplex PCR conditions in all subsequent assays with clinical samples.

### 3.3. Specificity of Multiplex Real-Time PCR Assays

A two-way amplification was carried out to measure primers and TaqMan probes specificities. In these assays, the amplification observed with each quadruplex PCR assay (Red, Orange, and Cariogenic complex) and gDNA from one bacterial species was comparable with the amplification with mixed genomic DNA. The comparison of Ct values in conjunction with curves shape suggested a lack of interactions between oligonucleotides and non-specific DNA targets (Appendix A). These PCR assays suggested the specificity of primers and TaqMan probes for their respective bacterial species and PCR conditions.

### 3.4. Testing of Clinical Samples

Because the number of analyzed bacteria differed significantly between individual samples, we used the log10-counts for the TB load and that of the 10 determined species within the multiplex PCR assays (Appendix A). The mean counts for the TB load and that of the 10 evaluated species in the interdental biofilm are shown in Figure 1. There was an excellent match between the concentration of TB found in the red and cariogenic complex quadruplex PCR assay (r = 0.97, *p* < 0.0001). 

The prevalence and percentage content of bacteria detected in samples from the control tooth and from the affected tooth are summarized in Table 3 and Appendix A. Whilst *P. micra* showed a tendency of higher prevalence in samples of plaque from the affected tooth, the percentage contents of *T. denticola*, *F. nucleatum*, and *A. actinomycetemcomitans* were significantly higher in samples from the affected tooth than from the control tooth (*p* < 0.05).

### 3.5. Validation of Method

Results obtained by real-time PCR assays correspond with the findings of the newly developed multiplex PCR method (Table 4). In addition, the possibility of false-positive results was further excluded by sequencing the selected positive samples (Appendix A). 

### 3.6. Correlation between the Counts of Bacteria and Their Percentage Content in Sample 

The total count of bacteria may span several orders of magnitude from one collection to another. Figure 2 shows the correlation between counts of bacteria and their percentage content in analyzed samples when we considered only those bacteria species positive in at least 10 samples. A significant correlation between the TB percentage and the count of bacteria was found only in the case of *F. nucleatum* (*p* = 0.02. *r* = 0.30).

## 4. Discussion

Nowadays, a variety of techniques have been used for the determination of microbial composition in the oral cavity. Methods based on the detection of specific bacterial DNA sequences have recently become invaluable in basic dental science and translational research due to their specificity, sensitivity, and rapid determination of specific oral pathogens. Compared to the other methods, real-time qPCR offers a sensitive means of detecting and quantifying a small number of bacteria in clinical samples.

In this study, three real-time multiplex PCR assays detecting and quantifying three bacterial species related to dental caries, three bacterial species of the most pathogenic periodontal red complex, and three bacterial species of the orange complex associated with periodontal disease together with *A. actinomycetemcomitans* were developed. In commercial DNA diagnostic tests, multiplex real-time PCR assays have become of great interest. The main advantages of these methods are the detection of more than one pathogen in a sample and significant saving of the qPCR reagent costs. On the other hand, in a multiplex real-time PCR, the amplification of multiple target sequences within a single reaction does not allow the use of ideal amplification conditions for each target. Despite slightly lower LODs for *T. denticola* and *Lactobacillus* spp. in simultaneous amplification of bacterial species in the multiplex reactions, the oligos demonstrate optimum amplification efficiency and reliability for all intended targets. Another limitation of the presented method might be the limited number of bacterial pathogens which can be investigated in a single assay. Currently, there are approximately 700 species listed in the human oral microbiome database (HOMD). Complete characterization of all these species utilizing the next-generation DNA sequencing methodologies, which are today an invaluable tool in basic research, is currently still expensive, laborious, and far from being accessible for routine sample analyses for most clinical laboratories.

Previously, in several studies, multiplex PCR was used as a tool for the detection of more than one periodontal or cariogenic pathogen in a sample [21,24]; however, these assays failed to provide accurate quantitative data. Only detecting the presence or absence of bacteria in a sample can be misleading, as these frequently occur in low numbers in healthy subjects, as confirmed in the current study. Recently, the multiplex real-time PCR assay for the detection and quantification of periodontal bacteria from clinical samples has been introduced [21]. However, this assay detects only several bacteria from the Socransky red and orange complexes associated with periodontal disease and does not include the determination of total bacterial (TB) content in the sample. The simple quantification of individual pathogenic bacteria in clinical samples can be misleading, because it does not involve any differences between individual sampling and DNA isolation method which can significantly influence the measured quantity of individual bacteria. This assumption was confirmed in this study: the correlation found between the quantity of TB and periodontal bacteria indicated the importance of TB determination in our assays for the determination of bacteria associated with periodontal disease (Socransky red and orange complexes). Moreover, the mean counts for the TB load and that of the ten evaluated species in the samples correspond very well to the previously published results [18,25,26,27].

In the tested samples from small children, healthy teeth with localization 73 or 83 were selected as the control for their highest efficiency of self-cleaning by saliva. This corresponds with the fact that primary incisors and canines of the lower jaw are least often affected by dental caries [28]. The significant difference between microbial composition obtained from the area of the control tooth and the affected tooth highlights the importance of standardization of sampling.

If risky bacterial species are identified, they can be eradicated/suppressed individually, either by conventional methods or by new biotechnological approaches, while allowing the recolonization of favorable and healthy microflora [29]. Analysis of the spectrum of microorganisms associated with the development of periodontal diseases can be an appropriate complement to the clinical and radiological examination of patients with periodontitis [30]. Tests can be used to more accurately diagnose periodontitis, especially when there are suspicions of aggressive forms in which we detect bacteria (*T. forsythia*, *P. gingivalis*, and *A. actinomycetemcomitans*) capable of penetrating into the cells of periodontal tissues in the subgingival space. These microorganisms are very difficult to mechanically eliminate from the periodontal throat (e.g., curettes, sonic instruments), so in the case of advanced periodontitis with an aggressive course, we can modify the treatment plan by adding antibiotics to the standard periodontological treatment [31,32].

However, the examination of subgingival microflora with the subsequent administration of antibiotics is only relevant in patients with good oral hygiene. Selected antibiotics are therefore exclusively administered together with the mechanical disruption of the microbial coating in the periodontal pocket during subgingival treatment or flap surgery. Separately applied antibiotics will only affect those bacteria that are able to penetrate into soft tissues, but those bound in the biofilm are extremely resistant to antimicrobial agents.

However, a number of experts have many reservations about the use of molecular biology methods in periodontology. One of these is testing only a limited number of locations, which can lead to distorted results. Therefore, it is recommended to look for sites with the deepest active periodontal pockets from each quadrant, where we expect the occurrence of microorganisms associated with severe forms of periodontitis. Another consideration is that many tests do not distinguish between invasive (pathogenic) and non-pathogenic serotype bacteria, which may lead to “overtreatment”. Often, the choice of a suitable antibiotic, which is not only indirect in culture, is empirically based on the detected spectrum of bacteria [32]. Unfortunately, not all oral bacteria can be cultured by conventional laboratory techniques [33].

Some authors also wonder whether antibiotics are necessary for the treatment of periodontitis [34,35], and whether in the treatment of periodontitis we focus too much on the elimination of bacteria that are a common part of the oral cavity microflora [35], especially when many of the periodontal bacteria can be found in people with healthy gingiva/periodontium and also in small children [36].

A recent meta-analysis was focused on the importance of antibiotics given during conservative periodontal treatment. The results confirmed that their additive effect is evident in patients younger than 55 years with severe periodontitis [31]. The authors recommend considering using antibiotics very individually and avoiding their flat-rate use. To do this, molecular biology testing of subgingival microflora could make a significant contribution to individual cases.

In our opinion, the performance of these microbial tests could be useful in the following cases: (i) when we suspect an aggressive form of periodontitis, (ii) when the initial instrumental phase of periodontal therapy did not have the predicted outcome and the periodontal disease does not respond to careful subgingival treatment for a patient with excellent oral hygiene, (iii) prior to flap surgery using controlled tissue regeneration in patients with advanced periodontitis, (iv) in extensive and demanding prosthetic or orthodontic treatment in patients with residual periodontal pockets, (v) in the preparation of implantological treatment in patients with periodontitis and implantological complications treatment (periimplantitis), and (vi) in patients with advanced periodontitis and an elevated health risk.

## 5. Conclusions

A simple, rapid, sensitive, reproducible, and economical method for the identification and quantification of selected cariogenic and periodontal bacteria was developed. Comprehensive microbiological examination showed that the risk of dental caries or periodontal disease may be assessed by this multiplex real-time PCR test despite some limitations, namely slightly lower sensitivity for some targets and a limited number of detected bacterial pathogens. The test could be also used as a tool for behavioral intervention in the context of oral hygiene and thus prevention of the aforementioned diseases. In clinical practice, the importance may also be in the gathering of supported data for the diagnosis and the verification of periodontal therapy. We also believe that the methodical approach to sample collection is very important.

## Figures and Tables

**Figure 1 diagnostics-10-00008-f001:**
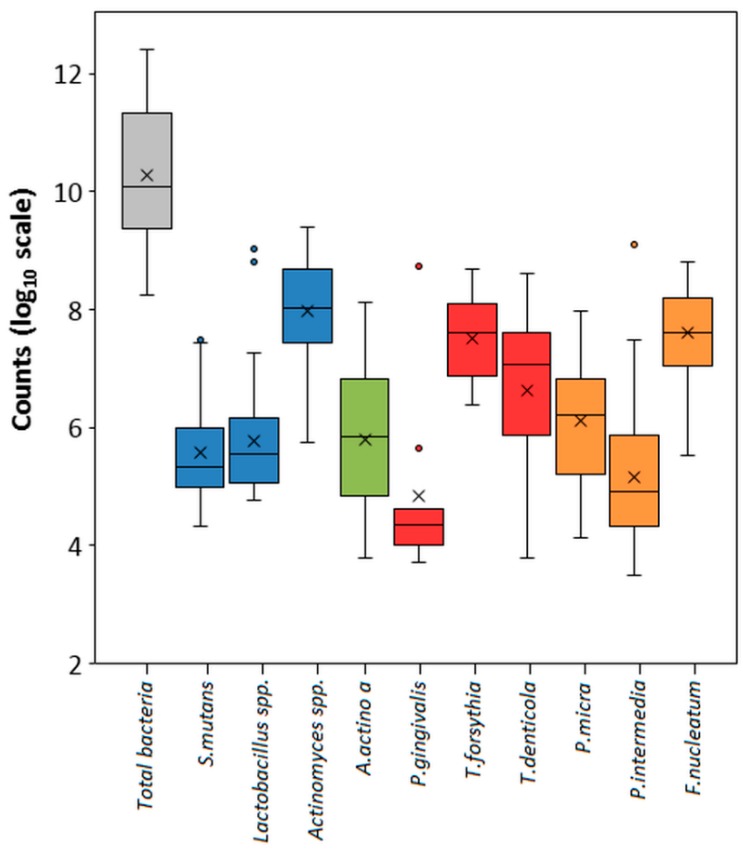
Abundance of bacterial species among the tested interdental site samples. The counts are reported on a log_10_ scale. Blue boxes correspond to the subtotals for species of cariogenic bacteria. Red and orange boxes correspond to the subtotals for species of the Socransky red and orange complexes.

**Figure 2 diagnostics-10-00008-f002:**
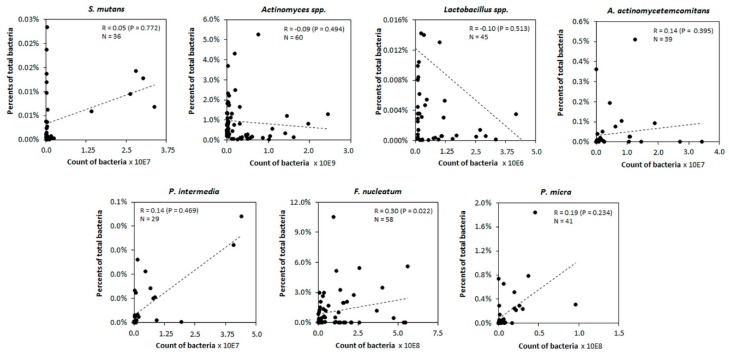
Correlation plots between the counts of bacteria and their percentage content in samples. Exceptional values in *Lactobacillus* spp. and *A. actinomycetemcomitans* were omitted from the analysis. Black dots correspond to individual analyzed samples and dotted line shows trend line on a scatter plot.

**Table 1 diagnostics-10-00008-t001:** Primers and TaqMan probes used in multiplex real-time assays.

Bacterial Strain	Target Gene	Sequence of Primers or Probe	Fluorophore	Amplicon Size (bp)	References
Cariogenic Pathogens and Total Bacteria Content					
*S. mutans*	gtfB	Forward: CCTACAGCTCAGAGATGCTAT		113	This study
		Reverse: GCCATACACCACTCATGAATT			This study
		Probe: TGGAAATGACGGTCGCCGTTAT	FAM		This study
*Lactobacillus* spp.	16S	Forward: TGGAAACAGGTGCTAATACC		232	[18]
		Reverse: GTCCATTGTGGAAGATTCCC			[18]
		Probe: CCACATTGGGACTGAGACACGG	HEX		This study
*Actinomyces* spp.	16S	Forward: CCTCTGGCTTAACTGGGG		88	[19]
		Reverse: CATTCCACCGCTACACCA			[19]
		Probe: TCCAGTCTCCCCTACCGCAC	Cy5		This study
Total bacteria content	16S	Forward: TGGAGCATGTGGTTTAATTCGA		130	[20]
		Reverse: ACGAGCTGACGACAACCATG			This study
		Probe: TGG+TAAGG+TTCTTCGCGT	Texas Red		This study
Orange Complex and *A. actinomycetemcomitans*					
*A. actinomycetemcomitans*	23S	Forward: GCGAAACGAAGAGAAGCAAG		111	[21]
		Reverse: CCTACCCAACAGGCGTATCA			[21]
		Probe: AAGTGCGGTTGGGAATTGAGGA	FAM		This study
*P. micra*	16S	Forward: AGTGGGATAGCCGTTGGAAA		100	[22]
		Reverse: GACGCGAGCCCTTCTTACAC			[22]
		Probe: ACCGCATGAGACCACAGAATCGC	Texas Red		This study
*P. intermedia*	16S	Forward: TCCACCGATGAATCTTTGGTC		103	[23]
		Reverse: ATCCAACCTTCCCTCCACTC			[23]
		Probe: CGTCAGATGCCATATGTGGACAAC	Cy5		This study
*F. nucleatum*	16S	Forward: GGCTTCCCCATCGGCATTCC		123	[21]
		Reverse: AATGCAGGGCTCAACTCTGT			[21]
		Probe: AGTTCCGCTTACCTCTCCAGTAC	HEX		This study
Red Complex and Total Bacteria Content					
*P. gingivalis*	23S	Forward: CTGCGTATCCGACATATC		134	[21]
		Reverse: GGTACTGGTTCACTATCG			[21]
		Probe: AGACATCCTGTGTGAATTGGCG	FAM		This study
*T. forsythia*	groL	Forward: GAGGTTGTGGAAGGTATG		108	[21]
		Reverse: GTAGATCAGAATGTACGGATT			[21]
		Probe: TCCGCTTATTTCGTGACCGAT	Cy5		This study
*T. denticola*	16S	Forward: GTTGTTCGGAATTATTGG		109	[21]
		Reverse: GATTCAAGTCAAGCAGTA			[21]
		Probe: AGGCGGTTAGGTAAGCCTG	HEX		This study
Total bacteria content	16S	Forward: TGGAGCATGTGGTTTAATTCGA		130	[20]
		Reverse: ACGAGCTGACGACAACCATG			This study
		Probe: TGG+TAAGG+TTCTTCGCGT	Texas Red		This study

FAM, Fluorescein amidites; HEX, Hexachloro-fluorescein; Cy5, Cyanine5; Texas Red, sulforhodamine 101 acid chloride.

**Table 2 diagnostics-10-00008-t002:** Limit of detection (LOD) determined by PROBIT regression analysis (90% confidence level), the efficiency of PCR reaction and R^2^ calibration fit values for quadruplex PCR assays.

Bacterial Strain	LOD90 (Log_10_cop./μL)	PCR Efficiency	Calibration Fit (R^2^)
*S. mutans*	3.8	0.93	1.00
*Lactobacillus* spp.	4.5	0.95	1.00
*Actinomyces* spp.	3.8	1.04	0.99
*A. actinomycetemcomitans*	3.0	1.08	0.99
*P. gingivalis*	3.5	1.11	0.99
*T. forsythia*	3.3	1.08	0.98
*T. denticola*	5.2	1.10	0.98
*P. micra*	2.9	1.07	1.00
*P. intermedia*	3.2	1.16	1.00
*F. nucleatum*	3.3	1.07	1.00

**Table 3 diagnostics-10-00008-t003:** The prevalence and percentage content of bacteria detected in the sample from the area of healthy teeth (HT) and affected teeth (AT).

	Prevalence			Percentage Content		
Bacterial Strain	AT (%)	HT (%)	*p* Value *	AT (%)	HT (%)	*p* Value ^†^
*S. mutans*	63.3	46.7	0.24	0.36	0.37	0.67
*Lactobacillus spp.*	83.3	83.3	0.22	0.36	0.40	0.33
*Actinomyces spp.*	100.0	100.0	0.90	0.57	0.63	0.87
*A. actinomycetemcomitans*	43.3	66.7	0.17	0.51	0.26	<0.01
*P. gingivalis*	6.7	0.0	0.60	0.49	0.29	0.44
*T. forsythia*	16.7	6.7	0.37	0.26	0.50	0.85
*T. denticola*	13.3	3.3	0.67	0.79	0.29	0.03
*P. micra*	70.0	50.0	0.09	0.29	0.49	0.44
*P. intermedia*	36.7	26.7	0.74	0.26	0.23	0.31
*F. nucleatum*	96.7	96.7	0.58	0.98	0.43	<0.01

* *p* values are calculated by chi-square statistic; † *p* values are calculated by Student *t*-test.

**Table 4 diagnostics-10-00008-t004:** Comparison of the tested and the reference method [6] for cariogenic and periodontal bacteria (red and orange complex and *A. actinomycetemcomitans*) determination in clinical samples.

Bacterial Strain	Really Positive	False-Positive	False-Negative	Really Negative
*S. mutans*	33	0	0	27
*Lactobacillus* spp.	50	0	0	10
*Actinomyces* spp.	60	0	0	0
*A. actinomycetemcomitans*	33	0	0	27
*P. gingivalis*	2	0	0	58
*T. forsythia*	7	0	0	53
*T. denticola*	5	0	0	55
*P. micra*	36	0	0	24
*P. intermedia*	19	0	0	41
*F. nucleatum*	58	0	0	2

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
