# Peer review of "Rapid Multiplex Real-Time PCR Method for the Detection and Quantification of Selected Cariogenic and Periodontal Bacteria"

_diagnostics, 2019, doi:10.3390/diagnostics10010008_

Round 1
Reviewer 1 Report
This is interesting work related with important scientific problem and it is concentrated on the new diagnostic method, so it is clearly within the scope of the “Diagnostic” journal.
The introduction is interesting and correct. The proposed aim is correct and adequate to the content. The methodology has been presented in appropriate way. The test results are clearly presented and properly described. The discussion is adequate to the results obtained. The conclusions are correct.
I have only few minor suggestions, which, however, are relevant to the order of work, but do not relate to its substantive value. So I marked "corrections to minor methodological errors and text editing" (minor revision).
The materials and methods section should be before the results section. In current for it is confusing. I suggest add scientific hypothesis in introduction section. Do the authors see limitations related to the proposed method? If so, maybe it is worth mentioning them in the discussion and conclusions.Author Response
Comment 1. The materials and methods section should be before the results section. In current for it is confusing.
Answer: We are sorry for this inconvenience, but the MS was written according to Instructions for Authors of Diagnostics.
https://www.mdpi.com/journal/diagnostics/instructions
Comment 2: I suggest add scientific hypothesis in introduction section.
Answer: In the Introduction section it is written: “There is an assumption that samples from children with sECC and gingivitis may be appropriate for the testing of a new sensitive method for oral bacterial screening. The main aim of this study was......”
Comment 3: Do the authors see limitations related to the proposed method? If so, maybe it is worth mentioning them in the discussion and conclusions.
Answer: Thank you for this critical comment, we have added the following sentence to discussion and conclussion section
Discussion, Line 200-217: In this study, three real-time multiplex PCR assays detecting and quantifying three bacterial species related to dental caries, three bacterial species of the most pathogenic periodontal red complex, and three bacterial species of the orange complex associated with periodontal disease together with A. actinomycetemcomitans were developed. In commercial DNA diagnostic tests, multiplex real-time PCR assays have become of great interest. Main advantages of these methods are the detection of more than one pathogen in a sample and significant saving of the qPCR reagent costs. On the other hand, in a multiplex real‐time PCR, the amplification of multiple target sequences within a single reaction does not allow the use of ideal amplification conditions for each target. Despite slightly lower LODs for T. denticola and Lactobacillus spp. in simultaneously amplification of bacterial species in the multiplex reactions, the oligos demonstrate optimum amplification efficiency and reliability for all intended targets. The another limitation of the presented method might be the limited number of bacterial pathogens which can be investigated in a single assay. Currently, there are approximately 700 species listed in the human oral microbiome database (HOMD). Complete characterization of all these species utilizing the next-generation DNA sequencing methodologies which are today an invaluable tool in basic research, is currently still expensive, laboured and far from being accessible for routine sample analyses for most clinical laboratories.
Conclusion, line 403-407: Comprehensive microbiological examination showed that the risk of dental caries or periodontal disease may be assessed by this multiplex real‐time PCR test despite some limitations, namely slightly lower sensitivity for some targets and limited number of detected pathogens.
Thank you very much for your comments and the positive evaluation of our article.
Reviewer 2 Report
Many methods were applied to detect bacterial antigen or specific DNA sequence. Among these methods,16S rRNA is the major target to determine the microbiome of dental caries. The previous methods could not coexisting high taxonomic resolution and economical classification. According to the previous studies, the bacterial groups form red complex and orange complex causing dental caries. A. actinomycetemcomitans as a potent immunoregulator of periodontal patients. In this study, the gnomic DNA from different groups of bacterial such as orange complex, red complex and cariogenic species are applied to quantify bacteria content by real-time PCR. The authors develop a novel method with the low LOD and low economic cost may apply for diagnostic process. Some minor problems need to corrected before publication.
The page numbers of references 5 and 27 should be recorded. According to the journal format, part two is material and methods. However, the material and method is part four in this manuscript. In line 108, the author refers that the high concentration of bacteria can avoid the lower LOD causing detection problem. In the figure1, the sample from gingivalis is the lowest concentration. Is the sample from the lowest concentration P. gingivalis can detect precisely? Which range of LOD90 in real-time PCR is detected from the previous studies with the same bacteria strain? In the table 3, the P values for some tests are >0.01. However, all the tests have P values > 0.01. Does “>0.01” actually represent “<0.01”?Author Response
Comment 1: The page numbers of references 5 and 27 should be recorded.
Answer: References were corrected.
Comment 2: According to the journal format, part two is material and methods. However, the material and method is part four in this manuscript.
Answer: MS was written according to the Instructions for Authors of Diagnostics, section Material and Methods should be as part four.
https://www.mdpi.com/journal/diagnostics/instructions
Comment 3: In line 108, the author refers that the high concentration of bacteria can avoid the lower LOD causing detection problem. In the figure 1, the sample from gingivalis is the lowest concentration. Is the sample from the lowest concentration P. gingivalis can detect precisely? Which range of LOD90 in real-time PCR is detected from the previous studies with the same bacteria strain?
Answer: It is true that P. gingivalis is present in the samples in the lowest concetration, however, due to 12 copies of rRNA gene in its genome and no false negative results in comparison with reference method precise detection should be expected.
In previous studies the LOD or LOQ range between 102 to 103 CFU (Carousel et al. 2016, Frontiers in Microbiology, Vol. 7; Coffey et al., 2016, Clin Exp Dent Res 2, 185–192; Sakamoto et al., 2001. Microbiol.Immun. 45, 39–44 ) and depends on the number of performed cycles. The lower LOD was achieved in studies with singleplex qPCR and 40 cycles (Carousel et al. 2016, Frontiers in Microbiology, Vol. 7; Sakamoto et al., 2001. Microbiol.Immun. 45, 39–44).
Comment 4: In the table 3, the P values for some tests are >0.01. However, all the tests have P values > 0.01. Does “>0.01” actually represent “<0.01”?
Answer:
Yes, there was a typing error in the table and in Table 3 we corrected P-values > 0.01 for P-values < 0.01.
Thank you again very much for your comments that as we hope helped improve quality of our article. MS was checked by our native English speaking colleague according to your suggestions.